# Novel Microsatellite Loci, Cross-Species Validation of Multiplex Assays, and By-Catch Mitochondrial Genomes on *Ochthebius* Beetles from Supratidal Rockpools

**DOI:** 10.3390/insects14110881

**Published:** 2023-11-15

**Authors:** Antonio José García-Meseguer, Adrián Villastrigo, Juana María Mirón-Gatón, Andrés Millán, Josefa Velasco, Irene Muñoz

**Affiliations:** 1Ecology and Hydrology Department, University of Murcia, 30100 Murcia, Spain; aj.garciameseguer@um.es (A.J.G.-M.); juanamaria.miron@um.es (J.M.M.-G.); acmillan@um.es (A.M.); 2Division of Entomology, SNSB-Zoologische Staatssammlung München, 81247 Munich, Germany; adrianvillastrigo@gmail.com; 3Department of Biodiversity, Ecology and Evolution, Complutense University of Madrid, 28040 Madrid, Spain; irenmu06@ucm.es

**Keywords:** water beetles, microsatellite markers, next-generation sequencing, cross-species validation, insect conservation

## Abstract

**Simple Summary:**

Supratidal rockpools stand out as some of the most extreme environments on the planet. These pools are exposed to the wave’s action, fluctuations in sea level, dramatic shifts in water temperature, high salinity levels, desiccation, and intense sunshine. As a result, only a handful of species can survive there, with aquatic beetles of the genus *Ochthebius* being the insects that dominate this kind of habitat on the Mediterranean coast. In this work, we genetically analysed four species (*O. lejolisii*, *O. subinteger*, *O. celatus*, and *O. quadricollis*) found along the Mediterranean coasts of Spain and Malta to develop microsatellite markers, which are repetitive DNA motifs, for the first time for this genus. Additionally, we obtained the complete mitochondrial genome of three species (*O. lejolisii*, *O. subinteger*, and *O. quadricollis*). These newly developed microsatellite markers and mitochondrial genomes for *Ochthebius* will be valuable in future studies for evolutionary and ecological research on the diversity of this genus: identification, genetic structure, population connectivity, etc.

**Abstract:**

Here we focus on designing, for the first time, microsatellite markers for evolutionary and ecological research on aquatic beetles from the genus *Ochthebius* (Coleoptera, Hydraenidae). Some of these non-model species, with high cryptic diversity, exclusively inhabit supratidal rockpools, extreme and highly dynamic habitats with important anthropogenic threats. We analysed 15 individuals of four species (*O. lejolisii*, *O. subinteger*, *O. celatus*, and *O. quadricollis*) across 10 localities from the Mediterranean coasts of Spain and Malta. Using next-generation sequencing technology, two libraries were constructed to interpret the species of the two subgenera present consistently (*Ochthebius* s. str., *O. quadricollis*; and *Cobalius*, the rest of the species). Finally, 20 markers (10 for each subgenus) were obtained and successfully tested by cross-validation in the four species under study. As a by-catch, we could retrieve the complete mitochondrial genomes of *O. lejolisii*, *O. quadricollis*, and *O. subinteger*. Interestingly, the mitochondrial genome of *O. quadricollis* exhibited high genetic variability compared to already published data. The novel SSR panels and mitochondrial genomes for *Ochthebius* will be valuable in future research on species identification, diversity, genetic structure, and population connectivity in highly dynamic and threatened habitats such as supratidal coastal rockpools.

## 1. Introduction

Understanding population structure and genetic diversity is of crucial concern for effective biodiversity conservation, as a population’s genetic makeup influences its responses to both natural and artificial selection. Thus, comprehending genetic patterns and the underlying genetic mechanisms driving variability is essential for determining the most suitable strategies for biodiversity management [1,2,3]. This information helps in identifying distinct genetic populations, establishing spatial boundaries, mapping genetic variability, and assessing dispersal capabilities and gene flow. It is particularly relevant in cases of increased susceptibility to extinction [4], such as populations or species with low genetic variability [2,5], narrow distribution [6], habitat loss, and/or fragmentation [7] or limited dispersal ability [8]. Consequently, conservation and restoration programs must consider genetic diversity to enhance species survival and successful long-term restoration [3]. However, these programs frequently prioritize vertebrates and plants, leaving invertebrates out of their scope, regardless of their greater diversity [9]. Invertebrates, particularly insects, confront threats and consequences that are currently a hot topic among conservation biologists because of the intensity of their decline [10], although it is an issue that is not without controversy and requires rigorous, balanced, and numerical analyses [11]. To effectively assess invertebrate biodiversity for conservation, it is crucial to identify species and differentiate them from other closely related and morphologically very similar ones [12,13]. Overlapping distribution ranges [14,15] and coexistence [16] further complicate species identification, especially when morphological convergence (but also morphological stasis, see [17]) obscure species identification by either external or internal characters (i.e., cryptic species). In such cases, molecular techniques are indispensable [18,19,20], even though this approach evidences additional challenges when working with non-model organisms. Moreover, molecular ecologists are increasingly in need of universal markers to study the mechanisms involved in patterns of diversity at the genetic and community levels [21], to test hypotheses relating to fine-scale spatiotemporal segregation, and for the estimation of demographic parameters [22].

To overcome the knowledge gap in non-model organisms’ conservation, such as invertebrates, there is an interesting approach involving a combination of affordable next-generation high-throughput sequencing (NGS) and advanced bioinformatics tools. This approach can be used to identify various potential molecular markers [23] that can effectively reveal genetic structure at both inter- and intrapopulation levels [3]. NGS techniques have demonstrated their usefulness in designing and characterizing sequence data for developing molecular markers. Among these markers, microsatellites, also known as short simple repeats, or SSRs, stand out as versatile and cost-effective options that do not require substantial investment in specialized equipment [24,25]. Microsatellites consist of polymorphic markers with tandem repeats of 1–6 nucleotides found throughout the genomes of both eukaryotic and prokaryotic species [26], which together with codominance and Mendelian inheritance, make them ideally suited for population genetics studies [23]. Typically, an SSR locus varies in length ranging from 5 to 40 repeats, with dinucleotide, trinucleotide, and tetranucleotide repeats being the most chosen options for molecular genetic studies [24]. The ongoing advances in NGS techniques are democratizing the discovery of SSR markers, making them accessible to an increasing number of research laboratories. Consequently, the development of custom SSR marker panels has become commonplace. SSR designs may be targeted to closely related species [27], making them potentially valuable tools for elucidating interspecific genetic relationships among recently diverged taxa. Furthermore, at smaller spatial scales and in more recent timeframes characterized by climate change and anthropogenic disturbances, SSR markers enable the study of range shifts and increased contact zones, especially in complex or dynamic habitats such as ecotones where hybridization processes between ecologically distinct species may occur.

At the boundary between marine and terrestrial realms, supralittoral coastal rockpools, above the high tide line, represent some of the most dynamic and extreme habitats on the planet. Supralittoral rockpools are characterized by extreme salinity concentrations [28,29], high daily thermal fluctuations [30], changes in sea level and wave action [31], and desiccation [32,33], among other stressors. These adverse conditions are so challenging that only adapted species can thrive there [34,35]. In this context, the very small (1–2 mm) aquatic beetles of the worldwide distributed genus *Ochthebius* Leach, 1815 (Coleoptera: Hydraenidae) are capable of inhabiting these extreme environments, with some of them exhibiting cryptic diversity [36] and even partial sympatry with overlapping niches [16]. However, *Ochthebius* species are absent in the intertidal zone, where the area is regularly covered by water with periods of circadian, circatidal, or circalunadian rhythms, and its organisms have a series of adaptations to it (in addition to physiological ones, they can also present avoidance mechanisms, fixation structures, and mechanical resistance) [37,38].

Mapping distribution ranges and population boundaries in insects is a difficult task [39,40], particularly within cryptic species groups like the mentioned Mediterranean supralittoral species of the genus *Ochthebius* [35,36,41,42,43]. In addition, many of these habitats face serious threats due to land use changes, such as coastal development [44], resulting in fragmentation and/or habitat loss that negatively impacts population connectivity and viability. Recent studies have used a combination of mitochondrial and nuclear markers to identify morphologically cryptic lineages within recognized taxa in the eastern Atlantic and western Mediterranean [36,42]. However, achieving finer resolution is essential to validate potential lineages and to fully understand their connectivity patterns among populations [40].

While recent studies have significantly improved our knowledge of genetic diversity within the genus *Ochthebius* [36,42,45], there are still gaps in our understanding of gene flow, hybridization, and population structure. Here, we used NGS technology to screen the *Ochthebius* genome with a primary focus on developing and characterizing SSR loci for population identification and genetic structure studies. Additionally, we applied bioinformatic tools to assemble genomic data and detect the mitochondrial genome of the studied species. This mitochondrial data offers complementary information to address the causes shaping population structure at larger time scales. Furthermore, it serves as a valuable resource for the identification of cryptic species, a subject of particular relevance within the genus *Ochthebius*. We tested these novel SSR loci for cross-species amplification in four *Ochthebius* species, three within the subgenera *Cobalius* (*O. lejolisii* Mulsant & Rey, 1861, *O. subinteger* Mulsant & Rey, 1861, and *O. celatus* Jäch, 1989) and one within *Ochthebius* s.str. (*O. quadricollis* Mulsant, 1844).

## 2. Materials and Methods

### 2.1. Microsatellite Design

Thirty-five adult individuals from four species of the *Ochthebius* genus (*Ochthebius lejolisii*, *O. subinteger*, and *O. quadricollis* from the Iberian Mediterranean coast, and *Ochthebius celatus* from Malta) were analysed (see Appendix A for site location information). Total genomic DNA was extracted using a DNeasy Blood & Tissue kit (Qiagen, Hildesheim, Germany), following the manufacturer’s protocol. DNA quantity and quality were determined by UV spectrophotometry using a NanoDrop 1000 Spectrophotometer, and all samples were standardized to a final concentration of 10 ng/μL.

To identify sequences containing a simple sequence repeat (SSR) motif, fifteen individuals (see Appendix A) and their extracted DNA were sent to Ecogenics Gmbh (Balgach, Switzerland) for microsatellite (SRR) development following an SRR enrichment protocol, SRR testing, and design of the multiplex Polymerase Chain Reactions (PCRs) for genotyping. For the search, design, and test of the SSR markers, we worked as follows: an Illumina TruSeq Nano library was built and sequenced for *O. lejolisii* on the Illumina MiSeq platform (Illumina, San Diego, CA, USA) using a nano v2 500 cycles sequencing chip. The resulting paired-end reads that passed Illumina’s chastity filter were subject to de-multiplexing and trimming of Illumina adaptor residuals. Then, FastQC v0.117 (Babraham Institute, Cambridge, UK) [46] was used as a quality-control tool. The paired-end reads were subsequently merged with the software USEARCH v10.0.240 [47] to, in silico, reform the sequenced molecule. The resulting merged reads were screened with the software Tandem Repeats Finder, v4.09 [48]. Primer pairs were then designed using Primer3 [49] for the SSR candidates using standard default values. Given the inconsistent performance of the obtained SRR for *O. quadricollis* (see below and Appendix A), an additional library was prepared for a pooled DNA sample of *O. quadricollis* and *O. subinteger*, following the same pipeline as outlined for the previous library.

### 2.2. Multiplex Optimization

From each library, we identified potential SSR candidates and subsequently optimized and characterized these candidates to assess their suitability for cross-species amplification. Loci were chosen for amplification testing based on the number of repeats (≥ 7), the product size (≥ 100 bp), the absence of primer-dimer, and the primer alignment score to the target sequence. Subsequently, these markers were tested for amplification in fifteen individuals representing the four species of the genus *Ochthebius* here studied (see Appendix A). PCRs for SSR amplification were performed in a final volume of 10 μL, including 2–10 ng of DNA, 1× HOT FIREPol MulitPlex Mix (Solis BioDyne), and 0.3 μM of each primer, with the forward primer labelled with an FAM dye. The PCR program comprised an initial denaturation step of 95 °C for 12 min, followed by 35 cycles with denaturation at 95 °C for 20 s, annealing at 60 °C (unless stated otherwise in Appendix A) for 50 s, extension at 72 °C for 120 s, and concluded with a final extension step at 72 °C for 5 min. Allele-calling was based on the observed pattern using an Applied Biosystems 3730 Sequencer with a GeneScan 500 LIZ Size Standard (Applied Biosystems, Waltham, MA, USA).

To facilitate future population-based studies, we carefully selected optimized loci based on their allele size ranges and the absence of apparent null alleles across the individuals. These optimized loci were then combined into multiplex PCRs panels, considering their allele size and primer annealing temperature (Table 1). Individuals were genotyped by assessing the allele size using forward primers labelled with fluorescent dyes (FAM, ATTO532, ATTO550, ATTO565). PCRs were conducted using Phire Animal Tissue Direct Kit (Thermo Scientific, Waltham, MA, USA) and following the manufacturer’s instructions. Multiplex PCRs were performed in a final volume of 8 μL that contained 1× Phire Tissue Direct PCR Master Mix with 1.5 mM MgCl_2_, 0.3–0.5 μM of each primer with the forward primer labelled (details in Table 1), 1.2 μM BSA (bovine serum albumin) and 2 μL of DNA extract (2–10 ng/μL). The PCR program comprised an initial denaturation step of 98 °C for 5 min, followed by 40 cycles with denaturation at 98 °C for 5 s, annealing at 60 °C, extension at 72 °C for 20 s, and a final extension step at 72 °C for 30 min. PCR products were visualized on a 1.5% agarose gel stained with GelRed (Biotium Inc., Fremont, CA, USA) and were sent to Secugen S.L. (Madrid, Spain) for fragment analysis using an Applied Biosystems 3730 Sequencer. GeneScan 500 LIZ (Applied Biosystems, Waltham, MA, USA) was used for accurate sizing. Allele sizes were scored and checked manually using GeneMapper software v.5 (Applied Biosystems, Waltham, MA, USA). All ambiguous peak profiles were considered as missing data.

Genetic diversity parameters based on SSR markers were estimated using GenAlEx 6.5 [50]. These parameters included the effective number of alleles (N_A_), number of effective alleles (N_EA_), and observed and expected levels of heterozygosity (H_O_ and H_E_, respectively). Principal component analysis (PCA) was performed using the “prcomp” function in R [51] to investigate genetic differentiation. Population structure analysis was carried out using Structure 2.3.4 software (Stanford University, California, USA) [52] with subgroups (K) set from 1 to 5. Each K was assessed through 10 independent runs. The project parameters encompassed a burn-in period of 100,000 iterations followed by 1,000,000 Monte Carlo Markov Chain (MCMC) replicates. The analyses assumed an admixture model and correlated allele frequencies. To determine the most likely number of populations (K), Structure results were imported into Structure Harvester [53] software to calculate ΔK [54]. The results of the runs for the best K were combined with CLUMPP software (version 1.1.2) (University of Michigan, Ann Arbor, MI, USA) [55]. The CLUMPP output files were visualized using StructuRly [56].

### 2.3. By-Catch Shotgun-Sequencing Data

Similar to other protocols for shotgun-sequencing, we anticipated the presence of potentially overrepresented sequences, specifically the mitochondrial genome, even though it was not the primary target (e.g., as in [57]). First, low-quality raw reads were discarded, and low-quality ends were trimmed using fastp v0.23.1 [58]. Next, SPAdes v3.15.5 [59] was used for library assembly. Contigs containing mitochondrial data were identified using *blast* [60] and already published mitochondrial genomes of *Ochthebius* [45]. Contigs were circularized searching for repetitive motifs with Geneious v10.2.6 [61], provisionally annotated using the Mitos 2 WebServer [62], and manually refined by comparison with reference genomes. Additionally, individual genes were extracted and aligned using MAFFT v7.450 [63], followed by the calculation of pairwise distance per gene within Geneious v10.2.6.

## 3. Results

### 3.1. Isolation and Characterization of Microsatellite Loci

A total of 2,035,605 raw reads were processed for *O. lejolisii* and 2,876,134 for *O. quadricollis*. The resulting merged reads were 91,615 for *O. lejolisii* and 99,953 for *O. quadricollis*. A total of 2650 (*O. lejolisii*) and 4037 (*O. quadricollis*) merged reads contained a microsatellite insert, for which primers could only be designed on 1276 and 1472 microsatellite fragments for *O. lejolisii* and *O. quadricollis,* respectively. For the selection of candidate markers, we used microsatellites with a tetra- or a trinucleotide of at least 6 repeat units or a dinucleotide of at least 10 repeat units, which generated 247 (*O. lejolisii*) and 394 (*O. quadricollis)* SSR candidates.

Among the candidate SSRs for *O. lejolisii*, dinucleotide repeats (68.42%) were more frequent than trinucleotide repeats (31.58%). Among the candidate SSRs of *O. quadricollis*, dinucleotide repeats (84.52%) were more frequent than tri- and tetranucleotide repeats (13.96% and 1.52% respectively).

The frequency distribution range of microsatellite repeats ranged from 11 to 30 repeats for dinucleotides, and from 7 to 12 repeats for trinucleotides among the candidate SSRs for *O. lejolisii* (Figure 1A). In the case of *O. quadricollis* SSR candidates, the frequency distribution range ranged from 11 to 30 repeats for dinucleotides, from 7 to 14 for trinucleotides, and from 7 to 9 for tetranucleotide repeats (Figure 1B).

In both cases, in the dinucleotide SSRs, AT and TA were the most abundant repeat types, accounting for 31.09% in *O. lejolisii* and 35.28 and 48.98%, respectively, in *O. quadricollis*. Of the trinucleotide repeats, TAA and TAC were the most abundant in *O. lejolisii* (5.04 and 5.88% respectively), while in *O. quadricollis,* the TTA repeat was the most frequent (3.55%) (Figure 1C).

Among these candidate SSRs, 143 loci were selected (48 for *O. lejolisii* and another 95 for *O. quadricollis*) for the amplification test following the criteria and protocols previously described. Details and sequences of the 143 selected loci are in Appendix A.

### 3.2. Multiplex Optimization and Cross-Species Amplification

The 48 SSR primers selected for *O. lejolisii* were tested for cross-species amplification on fifteen individuals belonging to the four sampled species of the genus *Ochthebius*. Twenty-four SSR primers showed no amplification, one was monomorphic (os_898357), and the other 23 SSR markers were polymorphic in the target species (Table 2), showing clear amplification profiles and reliable amplification in all the species tested except for *O. quadricollis* (Appendix A); therefore, it was necessary to build a specific library for this species.

For the 23 polymorphic primers, the allele number ranged from two to six, the effective allele number ranged from 1.00 to 5.00, and the observed and expected levels of heterozygosity varied from 0.00 to 1.00 and from 0.00 to 0.80, respectively (Table 2). When investigating these markers in each of the *Ochthebius* species, only six of them (os_37067, os_70525, os_712676, os_866755, os_1099692, and os_1225179) displayed alleles for all species, so together with three other primers that showed alleles for at least two of the species (os_120990, os_336684, and os_890215) and os_1099017, which worked well in *O. celatus,* they were selected to design multiplex PCR reactions for the *O. celatus*, *O. subinteger,* and *O. lejolisii* species set. The details of the designed multiplex PCR combinations are shown in Table 1 and the genetic diversity parameters of these 10 SSR markers by species are described in Table 2. PCA and Structure analysis showed that the 10 selected SSRs for multiplex PCR differentiate the three *Cobalius* species studied (Figure 2).

For the design of multiplex PCRs in *O. quadricollis*, it was necessary to test 95 candidate SSRs (Appendix A) obtained from the species-specific library. Forty-eight primer SSRs presented non-amplification or difficulties in interpretation for allele size selection, one was monomorphic (Oq_14408), and the remaining forty-seven were available for testing in seven *O. quadricollis* individuals (see Appendix A). For these 47 polymorphic SSRs, the allele number ranged from 2 to 6, the effective allele number ranged from 1.15 to 5.16, and the observed and expected levels of heterozygosity varied from 0.00 to 1.00 and from 0.13 to 0.81, respectively (Table 3). Considering the allele size ranges and the apparent lack of null alleles across the individuals tested, the Ecogenics company recommended 10 SSRs to design multiplex PCRs in the *O. quadricollis* species. The details of the designed multiplex PCR combinations are shown in Table 1 and the genetic diversity parameters of these 10 SSR markers are described in Table 3.

### 3.3. Mitochondrial Genomes

We recovered three mitochondrial genomes as “by-catch” from the raw data, which correspond with the species *O. quadricollis* (Figure 3), *O. lejolisii* (Figure 4), and *O. subinteger* (base coverage of mitochondrial contigs was 31×, 187×, and 18×, respectively; accession numbers *OR760222*, *OR760221,* and *OR760223,* respectively). These mitochondrial genomes include 13 protein-coding genes (PCGs), the mitochondrial small and large subunits of ribosomal RNA (rRNAs), and 22 transfer RNAs (tRNAs), 1 per amino acid except serine and leucine, which have two copies. All genes followed the standard gene order exhibited for Coleoptera. Due to the low-coverage data, the AT-rich control region situated between Ile-tRNA and the small subunit rRNA was not completely recovered for *O. subinteger*, and, thus, its mitochondrial genome was not circularized. In all cases, we detected incomplete stop codons in the following protein-coding genes: cytochrome *b*, cytochrome *c* oxidase subunits 1, 2, and 3, and NADH dehydrogenase subunits 2, 3, 4, and 5.

Pairwise distance comparison for the protein-coding genes and the ribosomal rRNAs has been done for *O. quadricollis* and *O. lejolisii*, for which reference mitochondrial genomes were available (accession numbers MT822975 and MT822977, respectively). We detected scarce genetic variability for *O. lejolisii*, with an average pairwise comparison of 0.18% (ranging between 0 and 0.63%). Comparison of *O. quadricollis* displays a high genetic variability, with an average distance of 8.34% (ranging between 5.55 and 9.94%) and 3.64% (ranging from 3.20 to 4.07%) for PCGs and rRNAs, respectively.

## 4. Discussion

Traditionally, designing microsatellites for non-model species was laborious and costly, involving marker isolation, cloning, and Sanger sequencing. NGS techniques have simplified the identification and characterization of microsatellites, establishing them as valuable markers in ecology and evolution for non-model species [64,65]. NGS techniques enabled the reliable discovery of microsatellite markers, a usual method for the description of de novo microsatellites applied in recent studies [66,67,68,69,70]. In this study, we successfully characterized new loci and mitochondrial genomes while performing optimization and cross-species validation of multiplex assays in four species of supralittoral *Ochthebius*. Of the 20 SSRs selected for multiplexing, 4 were dinucleotides, 15 were trinucleotides, and 1 was a tetranucleotide. Most of the tri- and tetranucleotide loci were preferred due to their reduced susceptibility to amplification errors, particularly stuttering [71], and their ease of scoring and rounding [72,73]. For dinucleotides, those with more than 10 repeat units were selected because higher repeat numbers have been linked to greater polymorphism resulting from polymerase slippage [74]. These criteria are important, especially for insects, which have been considered problematic for microsatellite isolation and genotyping [24,75]. Our data show that the 20 SSR loci selected are reliable markers for genetic analyses of individuals of the genus *Ochthebius*, offering excellent detection quality and high polymorphism.

Cross-species validation is decisive, particularly when full or partial genomic sequences are not available, as developing SSRs is time-consuming and costly [76,77,78,79]. The lack of highly polymorphic and easy-to-use molecular markers, such as SSRs, may be an important reason why genetic studies have not been carried out in certain groups [79]. Overall, the success of interspecies transfer is inversely related to the evolutionary distance separating the origin and focal species [76,79,80,81,82]. In the analysed species, differences emerged between those from the subgenera *Cobalius* and *Ochthebius*, with an estimated evolutionary divergence of c. 55 Mya [36]. Consequently, it was necessary to design two independent multiplex assays, one for the species of the genus *Cobalius* (*O. lejolisii*, *O. subinteger*, and *O. celatus*) and another for the species of the subgenus *Ochthebius* s.str. (*O. quadricollis*), formerly known as *Calobius* group [83]. Genetic structure analysis (PCA and Structure) identified three clusters within the subgenus *Cobalius*, which corresponds to the three species analysed in this genus (*O. lejolisii*, *O. subinteger*, and *O. celatus*).

The release of mitochondrial genome data significantly enhances our understanding of the target species, especially given the potential existence of cryptic lineages, as suggested by recent literature [35,36,42]. Mitochondrial genomes have already been published for several *Ochthebius* species [45], including two of the target species in this study: *O. lejolisii,* from a specimen of Ceuta (African coast of Spain) and *O. quadricollis,* from a specimen of Sicily (Italy). Comparing these genomes with those obtained from the Mediterranean coast of the Iberian Peninsula revealed minor differences in *O. lejolisii* (pairwise average of 0.18%) and significant genetic divergence in *O. quadricollis*, ranging from 3.2% to 9.94%. These results corroborate earlier findings [35,36,41,42,43,45,84,85], further supporting the presence of multiple cryptic lineages within coastal *Ochthebius* [86]. Additionally, the publication of the first mitochondrial genome for *O. subinteger* holds promise for further investigation into its potential crypticity within the western Palaearctic [36,42].

The design of the markers obtained in this work, along with the cross-species validation, enables population structure studies, genetic diversity assessment, and investigation into connectivity among *Ochthebius* subpopulations in a given area. Moreover, microsatellites open the possibility of much greater sensitivity in testing additional evolutionary questions than previous studies relying on a limited set of molecular markers. These questions may include topics such as sex-biased dispersal [87,88]. This will help to elucidate in the future much of the lack of knowledge about these organisms.

In summary, our study provides, for the first time, microsatellite markers for the aquatic beetle genus *Ochthebius* and presents complete mitochondrial genomes for three species (*O. quadricollis*, *O. lejolisii*, and *O. subinteger*). Our results demonstrate the usefulness of these markers for different species within the same genus, facilitating studies on population structure, genetic diversity, dispersal capacity, and connectivity. This research will enhance our understanding of the complex dynamics of these interesting organisms, adapted to one of the most inhospitable habitats on the planet.

## 5. Conclusions

The microsatellite markers presented in this study, along with the included mitochondrial genomes, represent a pioneering resource. They will prove highly beneficial for future research on *Ochthebius* aquatic beetles, which thrive in some of the most challenging habitats on the planet. With these tools, we can delve deeper into various aspects of this genus, such as identifying cryptic species, exploring their genetic diversity, investigating potential sex-biased dispersal, and assessing the genetic structure and connectivity of their populations.

## Figures and Tables

**Figure 1 insects-14-00881-f001:**
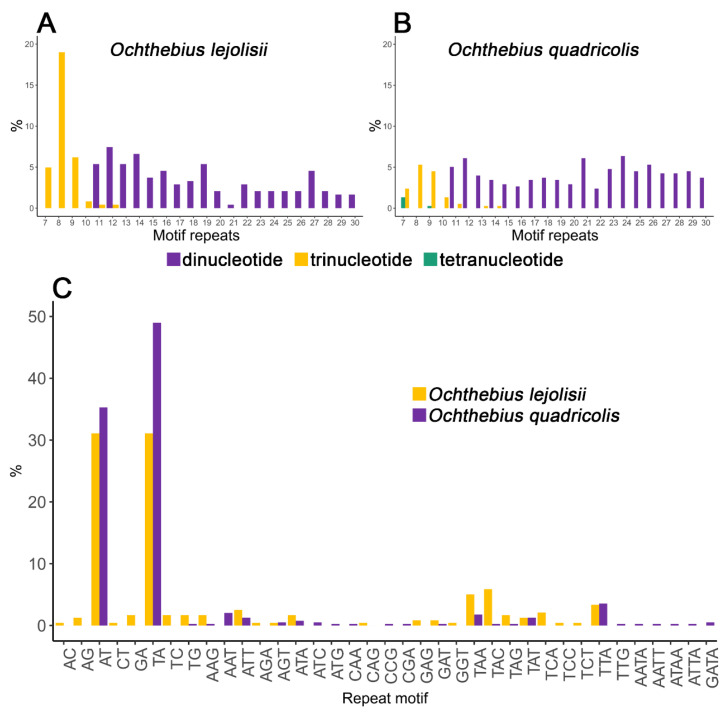
Frequency distribution by the number of repeats for dinucleotide, trinucleotide, and tetranucleotide in *O. lejolisii* (**A**) and *O. quadricollis* (**B**). Frequency of identified microsatellite motif types in *Ochthebius* species (**C**).

**Figure 2 insects-14-00881-f002:**
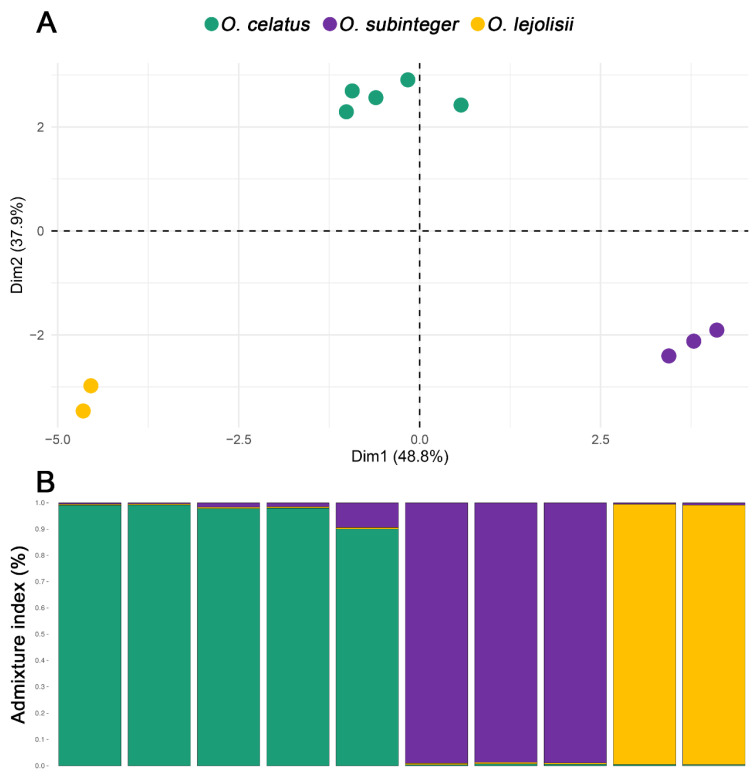
Principal component analysis (**A**) and cluster analysis using Structure software (**B**) in the subgenus *Cobalius* (*O. lejolisii*, *O. subinteger*, and *O. celatus*) based on the allelic variance of the 10 selected SSRs for multiplex PCR. Each dot represents an individual and each colour corresponds to the species assignment generated from the Structure analyses in K = 3.

**Figure 3 insects-14-00881-f003:**
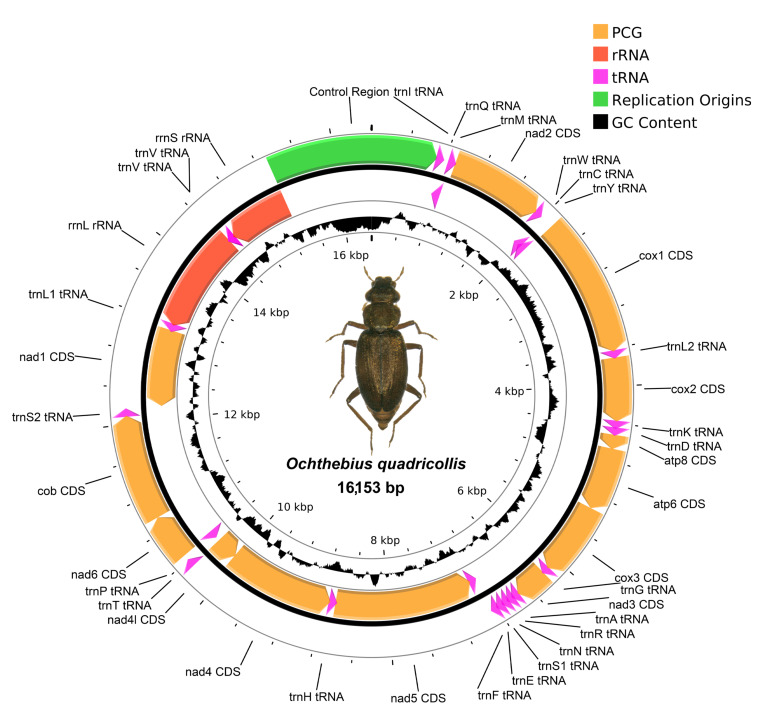
Visualization of the mitochondrial genome of *Ochthebius quadricollis*. Abbreviations: ATP6/ATP8: ATPase subunit 6/8, cox1–3: cytochrome *c* oxidase subunit 1–3, nad1–6/4l: NADH dehydrogenase subunits 1–6/4l, cob: cytochrome *b*, rrnL: large subunit of ribosomal RNA, and rrnS: small subunit of ribosomal RNA. All tRNA are labelled according to the IUPAC-IUB single-letter amino acid codes preceded by “trn”.

**Figure 4 insects-14-00881-f004:**
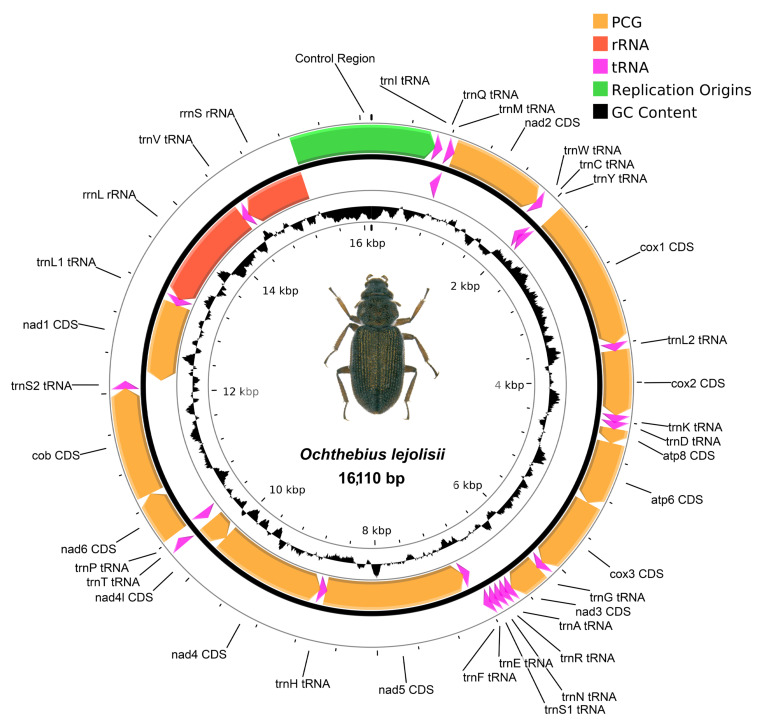
Visualization of the mitochondrial genome of *Ochthebius lejolisii*. Abbreviations: ATP6/ATP8: ATPase subunit 6/8, cox1–3: cytochrome *c* oxidase subunit 1–3, nad1–6/4l: NADH dehydrogenase subunits 1–6/4l, cob: cytochrome *b*, rrnL: large subunit of ribosomal RNA, and rrnS: small subunit of ribosomal RNA. All tRNA are labelled according to the IUPAC-IUB single-letter amino acid codes preceded by “trn”.

**Table 1 insects-14-00881-t001:** Locus characteristics at 20 novel optimized microsatellites combined in four multiplex PCR panels for *Ochthebius* species.

** *O. lejolisii, O. subinteger* ** ** and *O. celatus***									
**Locus**	**Primer Sequences 5′-3′**		**Motif**	**Allele Size** **Range (bp)**	**No. of** **Alleles**	**Multiplex**	**Dye**	**Final Primer** **Concentration (μM)**	**Annealing Temp. °C**
os_37067	F: R:	GGGAGCGGTGCATATTGTTG ACAAAGTGATAAAAAGCGAAAAGC	(TCT)_7_	220–241	4	1	FAM	0.3	60
os_120990	F: R:	TCGGAAAGGTGCTACTAACAAAC ATAATTGTCACTTGGACGACAG	(TA)_13_	135–208	5	1	ATTO550	0.3	60
os_890215	F: R:	CACAGGTCGGGGCTAAAATG TCGAAAACTTTAACCCAAGATTGC	(ATA)_8_	138–144	3	1	FAM	0.3	60
os_1099692	F: R:	TGCCACTTGCTCGAAGAAAC TCTCGTAAATTTTGTAGAGTTGGGG	(TTA)_8_	173–186	5	1	ATTO565	0.3	60
os_1225179	F: R:	AACAAAAGGCGCTTATGACG AGAACAATTACGTTCTACAATGTGC	(AT)_27_	125–130	2	1	ATTO532	0.3	60
os_70525	F: R:	ACAACAATCATGGAGGTCCG CGTAGGTCGAAAACTAATGTCCTC	(AAT)_9_	243–261	6	2	FAM	0.3	60
os_336684	F: R:	AGTTTCCTTACTTATCAAATAAAAGCG AGTCTGAAAAGCCCACTTGC	(AT)_24_	146–211	5	2	ATTO565	0.3	60
os_712676	F: R:	ATTACAGTGCGTCTGAGTGC AGACAACTTATTCCAACGAAGC	(AAT)_8_	85–97	4	2	ATTO532	0.3	60
os_866755	F: R:	CACCGATTGTATCAGCAGCC TGAACAAATAAAGTGCGCTTCTTC	(TAA)_8_	146–156	4	2	ATTO550	0.3	60
os_1099017	F: R:	AAATTAAAATTGGGATTTTCAAGTGC GGATGTGTATCAAAAATACTCTCTAGG	(TA)_25_	139–142	2	2	FAM	0.3	60
** *O. quadricollis* **									
**Locus**	**Primer Sequences 5′-3′**		**Motif**	**Allele Size** **Range (bp)**	**No. of** **Alleles**	**Multiplex**	**Dye**	**Final Primer** **Concentration (μM)**	**Annealing Temp. °C**
Oq_116301	F: R:	AGTCATGTTTGGTTAATGGATGTC ACTACAGTGAGTGACGTAAGC	(TTA)_9_	188–200	3	3	ATTO550	0.3	60
Oq_117536	F: R:	ACTCGGTGTTCCACAGATCG CATCAAGCCTTCTTCAGACCG	(TTA)_9_	207–217	3	3	FAM	0.3	60
Oq_1610909	F: R:	CGACCCTCTTCAATACCAAGC GTCCACCAAAGAACGAGGAC	(ATT)_9_	222–231	4	3	ATTO565	0.5	60
Oq_2015046	F: R:	TCCGTTTGAGAGTAGCACCC GGGACGGTATATGGGGATGG	(AGT)_10_	201–210	4	3	ATTO532	0.3	60
Oq_2184903	F: R:	ATGTTTGGACCGCCATTGTG TGTTAGTTTGATGATTTTCTTCGAC	(ATTA)_7_	118–143	2	3	ATTO565	0.5	60
Oq_122989	F: R:	ATAAATGGTGAGCAAGTAGCG ATATGGTACAACGGAGGCGG	(TAT)_8_	205–212	3	4	ATTO565	0.3	60
Oq_743481	F: R:	CACTCCAATTTGAACTACAATAAGTCC AGCATCCTCTGGTGATGTCC	(TAT)_8_	232–247	6	4	FAM	0.3	60
Oq_1061334	F: R:	TGTTTCCTAAGTGCTTGTGCG ACTGGTTACATTCAGCAAACTG	(TAA)_10_	170–228	5	4	ATTO550	0.5	60
Oq_1457302	F: R:	CTACATCCTGATCGGAGCCC CACCATCCAGAACACCAAGC	(ATA)_9_	180–194	6	4	FAM	0.3	60
Oq_2353236	F: R:	AACACTCCTAGTGCTCGCTC ATCTGGAGCTCATATCCGCC	(TTA)_8_	216–225	4	4	ATTO532	0.3	60

**Table 2 insects-14-00881-t002:** Genetic variation at 24 SSR primers selected for *O. lejolisii* and successfully tested for cross-species amplification in *O. celatus* and *O. subinteger*.

	*O. celatus*				*O. subinteger*				*O. lejolisii*				Total				
Locus	N_A_	N_E_	H_O_	H_E_	N_A_	N_E_	H_O_	H_E_	N_A_	N_E_	H_O_	H_E_	N	N_A_	N_E_	H_O_	H_E_
os_23931	0	0.000	0.000	0.000	2	1.385	0.333	0.278	0	0.000	0.000	0.000	3	2	1.385	0.333	0.278
os_37067 *	2	2.000	1.000	0.500	1	1.000	0.000	0.000	2	2.000	1.000	0.500	10	4	2.597	0.700	0.615
os_70525 *	4	2.778	0.800	0.640	2	1.385	0.333	0.278	1	1.000	0.000	0.000	10	6	4.545	0.500	0.780
os_120990 *	1	1.000	0.000	0.000	4	3.600	1.000	0.722	0	0.000	0.000	0.000	7	5	2.649	0.429	0.622
os_163989	0	0.000	0.000	0.000	2	1.385	0.333	0.278	0	0.000	0.000	0.000	3	2	1.385	0.333	0.278
os_166922	0	0.000	0.000	0.000	3	2.571	0.667	0.611	2	1.600	0.500	0.375	5	5	4.167	0.600	0.760
os_198491	0	0.000	0.000	0.000	4	3.600	0.333	0.722	0	0.000	0.000	0.000	3	4	3.600	0.333	0.722
os_258100	0	0.000	0.000	0.000	4	3.000	1.000	0.667	0	0.000	0.000	0.000	3	4	3.000	1.000	0.667
os_265699	0	0.000	0.000	0.000	4	3.600	0.667	0.722	2	1.600	0.500	0.375	5	6	5.000	0.600	0.800
os_314710	0	0.000	0.000	0.000	2	1.385	0.333	0.278	0	0.000	0.000	0.000	3	2	1.385	0.333	0.278
os_336684 *	1	1.000	0.000	0.000	4	3.600	0.333	0.722	0	0.000	0.000	0.000	6	5	3.130	0.167	0.681
os_458175	0	0.000	0.000	0.000	3	2.000	0.667	0.500	0	0.000	0.000	0.000	3	3	2.000	0.667	0.500
os_532182	0	0.000	0.000	0.000	3	2.571	1.000	0.611	0	0.000	0.000	0.000	3	3	2.571	1.000	0.611
os_537494	0	0.000	0.000	0.000	2	2.000	1.000	0.500	0	0.000	0.000	0.000	3	2	2.000	1.000	0.500
os_712676 *	1	1.000	0.000	0.000	2	1.800	0.667	0.444	1	1.000	0.000	0.000	10	3	1.852	0.200	0.460
os_858059	0	0.000	0.000	0.000	3	2.571	0.333	0.611	0	0.000	0.000	0.000	3	3	2.571	0.333	0.611
os_866755 *	2	1.220	0.200	0.180	2	1.385	0.333	0.278	2	1.600	0.500	0.375	10	4	2.740	0.300	0.635
os_890215 *	1	1.000	0.000	0.000	3	2.571	0.333	0.611	0	0.000	0.000	0.000	8	3	1.471	0.125	0.320
os_897615	0	0.000	0.000	0.000	2	2.000	1.000	0.500	1	1.000	0.000	0.000	4	2	1.882	0.750	0.469
os_898357	0	0.000	0.000	0.000	1	1.000	0.000	0.000	0	0.000	0.000	0.000	3	1	1.000	0.000	0.000
os_953020	0	0.000	0.000	0.000	3	2.667	1.000	0.625	1	1.000	0.000	0.000	3	4	3.600	0.667	0.722
os_1099017 *	2	1.923	0.800	0.480	0	0.000	0.000	0.000	0	0.000	0.000	0.000	5	2	1.923	0.800	0.480
os_1099692 *	3	1.515	0.400	0.340	3	3.000	1.000	0.667	1	1.000	0.000	0.000	10	5	3.922	0.500	0.745
os_1225179 *	1	1.000	0.000	0.000	1	1.000	0.000	0.000	1	1.000	0.000	0.000	10	2	1.724	0.000	0.420

Microsatellites marked with an asterisk (*) are those selected for multiplex PCR reactions in panels 1 and 2 (details in Table 1). N, number of individuals with reliable amplification; N_A_, number of alleles; N_E_, number of effective alleles; H_O_, observed heterozygosity; H_E_, expected heterozygosity.

**Table 3 insects-14-00881-t003:** Genetic variation at 47 SSR primers selected for *O. quadricollis*.

Locus	N	N_A_	N_E_	H_O_	H_E_
Oq_116301 *	7	3	1.342	0.286	0.255
Oq_117536 *	7	3	1.342	0.286	0.255
Oq_122989 *	7	3	2.800	0.714	0.643
Oq_129897	7	5	2.227	0.714	0.551
Oq_222765	7	3	2.000	0.714	0.500
Oq_235636	7	4	2.970	0.571	0.663
Oq_349914	7	2	1.508	0.429	0.337
Oq_467933	7	2	1.324	0.286	0.245
Oq_475499	7	6	4.083	0.857	0.755
Oq_556301	7	2	2.000	0.714	0.500
Oq_558655	7	2	1.690	0.000	0.408
Oq_673482	7	2	1.153	0.143	0.133
Oq_730507	7	5	3.379	0.571	0.704
Oq_743481 *	7	4	1.849	0.571	0.459
Oq_793953	7	2	1.153	0.143	0.133
Oq_801691	7	2	1.960	0.571	0.490
Oq_857784	7	2	1.508	0.429	0.337
Oq_958857	7	3	2.085	0.286	0.520
Oq_1061334 *	7	5	3.920	0.857	0.745
Oq_1151835	7	6	3.769	0.571	0.735
Oq_1217367	7	3	1.342	0.286	0.255
Oq_1227338	7	6	4.900	1.000	0.796
Oq_1237963	7	4	2.882	0.286	0.653
Oq_1388647	7	2	1.153	0.143	0.133
Oq_1457302 *	7	6	3.769	1.000	0.735
Oq_1489818	7	5	2.649	0.857	0.622
Oq_1493143	6	6	4.800	0.167	0.792
Oq_1555513	7	3	2.085	0.714	0.520
Oq_1557685	7	3	2.279	0.571	0.561
Oq_1610909 *	7	4	3.379	1.000	0.704
Oq_1708927	7	2	1.153	0.143	0.133
Oq_1714988	7	2	1.153	0.143	0.133
Oq_1750394	7	5	3.379	0.571	0.704
Oq_1763578	7	2	1.960	0.571	0.490
Oq_1821745	7	2	1.153	0.143	0.133
Oq_1879905	7	3	1.782	0.571	0.439
Oq_1966394	7	2	1.153	0.143	0.133
Oq_2006473	7	2	1.960	0.286	0.490
Oq_2010748	7	6	5.158	0.714	0.806
Oq_2015046 *	7	4	3.161	0.857	0.684
Oq_2044728	7	3	2.970	0.143	0.663
Oq_2057906	7	2	1.849	0.429	0.459
Oq_2144776	7	4	2.970	0.286	0.663
Oq_2145901	7	2	1.153	0.143	0.133
Oq_2184903 *	7	2	1.508	0.143	0.337
Oq_2254408	7	3	2.882	0.571	0.653
Oq_2353236 *	7	4	2.882	0.714	0.653

Microsatellites marked with an asterisk (*) are those selected for multiplex PCR reactions in panels 3 and 4 (details in Table 1). N, number of individuals with reliable amplification; N_A_, number of alleles; N_E_, number of effective alleles; H_O_, observed heterozygosity; H_E_, expected heterozygosity.

## Data Availability

All data produced in this study are available through the main text and the Appendix A.

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
