# Peer review of "Novel Microsatellite Loci, Cross-Species Validation of Multiplex Assays, and By-Catch Mitochondrial Genomes on *Ochthebius* Beetles from Supratidal Rockpools"

_insects, 2023, doi:10.3390/insects14110881_

Round 1

Reviewer 1 Report

Comments and Suggestions for Authors

The authors present a manuscript on the design of microsatellites using high-throughput sequencing for a genus of water beetles. Overall, the manuscript is well-written and convincing. I don’t have any serious criticism on the methods or on the presentation of the results. My main point is that it does not really become clear why the specific method of generating microsatellite markers using high-throughput sequencing was applied; specifically, what is the benefit of this approach vs. ‘simply’ using whole genome re-sequencing or something like ddRAD sequencing? To clarify this, the authors should present something like a plan for further studies. If they plan a full-scale phylogeographic study, I wonder if the approach presented here has any benefits over WGS / ddRAD. Sanger sequencing many microsatellite loci for many samples in one batch may not necessarily be cheaper than NGS (or will it be?). One advantage I can see is that new samples could be sequenced in small batches, by and by. But what is the advantage of this approach? This needs to be clarified.

Minor comments:

l. 14: This is the first time I hear that Ochthebius “dominates” all supratidal pools. I think there are a lot of other organisms, though Ochthebius may be the dominant among insects (!) in some areas.

l. 75: Explain why micorsatellites are useful: Especially for inclusion of low sample sizes. Compare to, e.g., ddRAD or other full NGS applications.

l. 95: It would be nice to have a very bried overview of supratidal rock pool organisms, as compared to tidal pools.

l. 107: Is this the global range of Ochthebius? Where (else) does the genus occur, and why did you sample where you sampled?

l. 135: I don’t really understand: There were 35 samples, but only 15 of these were Illumina sequenced. From which of these samples are the mitogenomes? And did you use the remaining 20 samples for PCR / Sanger sequencing?

l. 136: “we worked with two independent approaches”: I don’t understand which two approaches there are. There are two libraries, constructed with the same pipeline?

l. 366: “Much greater sensitivity”: Much greater than what?

Reference list: Check for italics of species names.

Kind regards,

Author Response

The authors present a manuscript on the design of microsatellites using high-throughput sequencing for a genus of water beetles. Overall, the manuscript is well-written and convincing. I don’t have any serious criticism on the methods or on the presentation of the results. My main point is that it does not really become clear why the specific method of generating microsatellite markers using high-throughput sequencing was applied; specifically, what is the benefit of this approach vs. ‘simply’ using whole genome re-sequencing or something like ddRAD sequencing? To clarify this, the authors should present something like a plan for further studies. If they plan a full-scale phylogeographic study, I wonder if the approach presented here has any benefits over WGS / ddRAD. Sanger sequencing many microsatellite loci for many samples in one batch may not necessarily be cheaper than NGS (or will it be?). One advantage I can see is that new samples could be sequenced in small batches, by and by. But what is the advantage of this approach? This needs to be clarified.

Thank you very much for your comment. Indeed, sequencing technologies are becoming more accessible, and selecting the right methodology can be challenging. We understand that each sequencing method comes with its own set of strengths and limitations, and the choice of methodology should align closely with the specific research objectives and constraints and the expertise of the research team. In our case, the absence of a reference genome for any Ochthebius’ species led us to refrain from using a WGS or ddRAD approach. Additionally, our decision to utilise microsatellites was heavily influenced by our team’s extensive expertise in this methodology. This familiarity not only streamlined the experimental process but allowed us to interpret the results with confidence and accuracy.

Generally, genotyping experiments very much depend on the organism and the amount of knowledge around it. Microsatellite genotyping is much more cost-effective per tested individual than sequencing based methods. Those methods also require initial establishment (e.g. enzyme selection for RAD approaches) and come along with a tremendous demand for bioinformatics analyses for the actual genotype data evaluation step. High-throughput methodology is commonly used for the de novo identification of microsatellite motives in an organism for which no published markers are available or for which the development of a novel set is desired. For this, we do not apply reduced representation sequencing methods such as ddRAD, because they target only a fraction of the genome and hence, the number of microsatellites that can potentially be detected is lower than in whole-genome sequencing.

Minor comments:

  1. 14: This is the first time I hear that Ochthebius “dominates” all supratidal pools. I think there are a lot of other organisms, though Ochthebius may be the dominant among insects (!) in some areas.

Thank you very much for letting us clarify this sentence. Our intention was to highlight that Ochthebius represent the dominant insect taxa in this ecosystem on the Mediterranean coast.

Please see now L14-15:

As a result, only a handful of species can survive there, with aquatic beetles of the genus Ochthebius being the insects that dominate this kind of habitat on the Mediterranean coast.

  1. 75: Explain why micorsatellites are useful: Especially for inclusion of low sample sizes. Compare to, e.g., ddRAD or other full NGS applications.

Microsatellites may continue to be used for a wide variety of population or conservation questions, despite the extensive development of genomic techniques in recent years, as previous studies indicate that microsatellite data can infer similar biological processes and patterns (Muñoz et al. 2017, Henriques et al. 2018, Hauser et al. 2021). In populations with low genetic diversity or that have experienced bottlenecks, multi-allelic markers, such as microsatellites, could provide the power needed in parentage analyses where SNPs cannot, and the costs for studying large numbers of samples and monitoring populations over time and on a large spatial scale are lower for microsatellites. In our case, the design of these microsatellite markers will be used in future studies with large numbers of individuals. These markers have the characteristic of being transferable between related species of the same genus, opening the possibility of studying the same locus in different species.

  1. 95: It would be nice to have a very bried overview of supratidal rock pool organisms, as compared to tidal pools.

Done as suggested. Please see now L102-106:

However, Ochthebius species are absent in the intertidal zone, where the area is regularly covered by water with periods of circadian, circatidal or circalunadian rhythms, and its organisms have a series of adaptations to it (in addition to physiological ones, they can also present avoidance mechanisms, fixation structures and mechanical resistance) [37,38].

  1. 107: Is this the global range of Ochthebius? Where (else) does the genus occur, and why did you sample where you sampled?

Ochthebius beetles are worldwide distributed, but recent systematic revisions of this genus have revealed specific subgenera and species groups that are restricted to supratidal rockpools in several regions (Villastrigo et al., 2019; 2020). The most speciose lineages inhabiting rockpools are found in the western Mediterranean, where there is evidence of multiple cryptic lineages. To investigate this further, our sampling was concentrated on the Mediterranean coast of the Iberian Peninsula, as recent studies have indicated the presence of cryptic diversity (Villastrigo et al., 2020, 2022). We designed microsatellites to be compatible not only with the Iberian fauna but also with another species found in the central and eastern Mediterranean (Ochthebius celatus).

However, we do appreciate your comment and to help clarify as suggested we have added the following (L99):

In this context, the very small (1-2 mm) aquatic beetles of the worldwide distributed genus Ochthebius Leach, 1815 (Coleoptera: Hydraenidae) …

  1. 135: I don’t really understand: There were 35 samples, but only 15 of these were Illumina sequenced. From which of these samples are the mitogenomes? And did you use the remaining 20 samples for PCR / Sanger sequencing?

Thanks for your comment. To make the libraries from which the mitogenomes and all candidate microsatellites were obtained, the samples now indicated in Table S1 with "Mitochondrial genome" were used. The other samples were used to test the multiplexes, as the individuals must be different to test their usefulness. We have also corrected an error in this table: in Cala Reona, it was O. subinteger, instead of O. lejolisii.

See now the corrected Table S1.

  1. 136: “we worked with two independent approaches”: I don’t understand which two approaches there are. There are two libraries, constructed with the same pipeline?

Thank you for this appreciation. We have clarified the requested sentence for a better understanding. Please see now L143.

  1. 366: “Much greater sensitivity”: Much greater than what?

We appreciate your comment: In that sentence, we aimed to emphasize that microsatellites offer a superior and more robust source of information for examining evolutionary questions compared to previous studies that relied on a limited set of genetic markers (i.e., COI and wingless genes).

Please see now L373-375:

Moreover, microsatellites open the possibility of much greater sensitivity in testing additional evolutionary questions than previous studies relying on a limited set of molecular markers. These questions may include topics such as sex-biased dispersal.

 Reference list: Check for italics of species names.

Done.

--

References:

Hauser, S. S., Athrey, G., & Leberg, P. L. (2021). Waste not, want not: Microsatellites remain an economical and informative technology for conservation genetics. Ecology and Evolution, 11(22), 15800-15814.

Henriques, D., Browne, K. A., Barnett, M. W., Parejo, M., Kryger, P., Freeman, T. C., ... & Pinto, M. A. (2018). High sample throughput genotyping for estimating C-lineage introgression in the dark honeybee: an accurate and cost-effective SNP-based tool. Scientific reports, 8(1), 8552.

Muñoz, I., Henriques, D., Jara, L., Johnston, J. S., Chávez‐Galarza, J., De La Rua, P., & Pinto, M. A. (2017). SNPs selected by information content outperform randomly selected microsatellite loci for delineating genetic identification and introgression in the endangered dark European honeybee (Apis mellifera mellifera). Molecular ecology resources, 17(4), 783-795.

Villastrigo, A., Jäch, M. A., Cardoso, A., Valladares, L. F., & Ribera, I. (2019). A molecular phylogeny of the tribe Ochthebiini (Coleoptera, Hydraenidae, Ochthebiinae). Systematic Entomology, 44(2), 273-288.

Villastrigo, A., Hernando, C., Millán, A., & Ribera, I. (2020). The neglected diversity of the Ochthebius fauna from Eastern Atlantic and Central and Western Mediterranean coastal rockpools (Coleoptera, Hydraenidae). Organisms Diversity & Evolution, 20, 785-801.

Villastrigo, A., Bilton, D. T., Abellán, P., Millán, A., Ribera, I., & Velasco, J. (2022). Cryptic lineages, cryptic barriers: historical seascapes and oceanic fronts drive genetic diversity in supralittoral rockpool beetles (Coleoptera: Hydraenidae). Zoological Journal of the Linnean Society, 196(2), 740-756.

Reviewer 2 Report

Comments and Suggestions for Authors

This is very interesting article with the excelent general introduction, very well writen. In my opinion both significance and the scientific soundness would be improved by giving the discussion wider perspective (see comment to lines 363-367). 

Specific comments:

line 21: please specify what kind of connectivity ( Authors prabably consider genetic connectivity

line 36: please specify what kind of connectivity ( Authors prabably consider genetic connectivity

line 125: specify if all adult individuals

line 247-248: Please comment in the discussion, it seems that to build a specific library for some species contradicts the usefullnes of the approach as a valuable tool

lines 335-336: Can you relate this statment to the other Coleoptera?

lines 363-367: Pleasae provide broader perspective in the discussion, bring some examples for higher taxonomic levels ( Hydraenidae? Coleoptera?)

Author Response

This is very interesting article with the excelent general introduction, very well writen. In my opinion both significance and the scientific soundness would be improved by giving the discussion wider perspective (see comment to lines 363-367).

Thank you for your positive evaluation and also for your comment concerning expanding our discussion. Unfortunately, our molecular information is restricted to the Mediterranean rockpools limiting its extrapolation.

Specific comments:

line 21: please specify what kind of connectivity ( Authors prabably consider genetic connectivity

Done as suggested. Please see now L21-22:

… ecological research on the diversity of this genus: identification, genetic structure, population connectivity, etc.

line 36: please specify what kind of connectivity ( Authors prabably consider genetic connectivity

Done as suggested. Please see now L36:

... , genetic structure, and population connectivity in highly dynamic and threatened habitats such as supratidal coastal rockpools.

line 125: specify if all adult individuals

Done as suggested. Please see now L132:

Thirty-five adult individuals from four species of the Ochthebius genus ...

line 247-248: Please comment in the discussion, it seems that to build a specific library for some species contradicts the usefullnes of the approach as a valuable tool

Two libraries were necessary, as we addressed the design of markers for two subgenera within the Ochthebius, with a significant evolutionary divergence (estimated at 55 Mya), subgenus Ochthebius s. str. and subgenus Cobalius. The specificity of this technique is high, but the design of these markers for the two mentioned subgenera will allow the study of Ochthebius from Mediterranean rockpools.

lines 335-336: Can you relate this statment to the other Coleoptera?

Microsatellite marker techniques, and molecular techniques in general, must be of high specificity to be able to carry out studies on a fine taxonomic resolution. Therefore, extending the validity of their usefulness to the Family or Order level is not adequate and is not the aim of this work either. For working at higher hierarchical levels, other molecular techniques may be more accurate (e.g., ultraconserved elements) and thus, are not commonly addressed using microsatellites. We focused especially on the Ochthebius of the Mediterranean rockpools.

lines 363-367: Pleasae provide broader perspective in the discussion, bring some examples for higher taxonomic levels ( Hydraenidae? Coleoptera?)

Please, see the comments above.

Reviewer 3 Report

Comments and Suggestions for Authors

The presented article by Garcia-Meseguer et al. discusses the development of a new set of microsatellite markers for the beetles of the genus Ochthebius. These markers are expected to be highly useful in conducting population genetic studies of these organisms. The article emphasizes the importance of developing new molecular markers for non-model species, rather than solely relying on obtaining genome sequences by NGS, which has become increasingly popular in recent years.

Author Response

The presented article by Garcia-Meseguer et al. discusses the development of a new set of microsatellite markers for the beetles of the genus Ochthebius. These markers are expected to be highly useful in conducting population genetic studies of these organisms. The article emphasizes the importance of developing new molecular markers for non-model species, rather than solely relying on obtaining genome sequences by NGS, which has become increasingly popular in recent years. 

Thank you very much for your positive evaluation.